# Dietary Protein Patterns during Pregnancy Are Associated with Risk of Gestational Diabetes Mellitus in Chinese Pregnant Women

**DOI:** 10.3390/nu14081623

**Published:** 2022-04-13

**Authors:** Weijia Wu, Nu Tang, Jingjing Zeng, Jin Jing, Li Cai

**Affiliations:** 1Department of Maternal and Child Health, School of Public Health, Sun Yat-sen University, Guangzhou 510080, China; wuwj28@mail.sysu.edu.cn (W.W.); jingjin@mail.sysu.edu.cn (J.J.); 2Department of Scientific Research, Hainan Women and Children’s Medical Center, Haikou 570206, China; 3Department of Health Care, Foshan Women and Children Hospital, Foshan 528000, China; tangn532@163.com; 4Evidence-Based Medicine Centre, Office of Academic Research, Xiangyang Central Hospital, Affiliated Hospital of Hubei University of Arts and Science, Xiangyang 441000, China; zengjj9@mail2.sysu.edu.cn; 5Guangdong Provincial Key Laboratory of Food, Nutrition and Health, School of Public Health, Sun Yat-sen University, Guangzhou 510080, China

**Keywords:** protein, dietary pattern, gestational diabetes mellitus

## Abstract

Controversies around the association between dietary protein intake and gestational diabetes mellitus (GDM) persist. To the best of our knowledge, this association has not previously been reported from the perspective of dietary protein patterns. We aimed to investigate the relationship between dietary protein patterns and GDM risk in pregnant women, and 1014 pregnant women (20–28 weeks of gestation) were recruited in Guangzhou, China, during 2017–2018. Maternal dietary information was collected by a validated food frequency questionnaire, which covered the most common foods consumed in Guangzhou, China. GDM was identified by a 75g oral glucose tolerance test. A K-means cluster analysis was conducted to aggregate individuals into three groups, which were determined by the major sources of protein. Logistic regression was employed to explore the relationship between dietary protein patterns and the risk of GDM. Among the 1014 participants, 191 (18.84%) were diagnosed with GDM. In the total population, when comparing the highest quartile with the lowest, we found that total protein and animal protein intake increased the risk of GDM with the adjusted odds ratios (95%CI) being 6.27, 5.43 (1.71–23.03, 1.71–17.22), respectively. Pregnant women were further divided into three dietary protein patterns, namely, white meat, plant–dairy–eggs, and red meat protein patterns. Compared to women with the plant–dairy–eggs protein pattern, those with the red meat protein pattern (OR: 1.80; 95%CI: 1.06–3.07) or white meat protein pattern (OR: 1.83; 95%CI: 1.04–3.24) had an increased risk of GDM. Higher dietary intakes of total or animal protein during mid-pregnancy were related to an increased risk of GDM. Furthermore, we first found that, compared to women with the plant–dairy–eggs protein pattern, women with the red meat or white meat protein patterns had a higher risk of GDM.

## 1. Introduction

Gestational diabetes mellitus (GDM) is defined as any degree of glucose intolerance with onset or first recognition during pregnancy [1,2]. Considerable evidence shows that women with GDM have a substantially increased short-term risk of adverse perinatal effects [3] and long-term risk of cardiovascular diseases and developing type 2 diabetes mellitus (T2DM) [4,5]. Additionally, the prevalence of GDM has dramatically increased over the past decades globally, especially in mainland China, ranging from 5% to 24% [6,7]. Collectively, it is imperative to identify the modifiable factors for GDM prevention, among which nutritional strategies play an essential role [8,9].

Dietary protein and amino acids are essential modulators of glucose metabolism, and a high intake of protein has detrimental effects on glucose homeostasis by promoting insulin resistance and increasing gluconeogenesis [10]. Dietary protein has been shown to involve in the development of GDM [11]. However, the association between dietary protein intakes and GDM risks still controversial. Several observational studies have reported that animal protein intake was positively related to an increased risk of GDM [12,13,14,15,16], whereas another study reported a non-significant relationship [17]. On the other hand, previous studies have shown a negative association between plant-based protein and GDM risk [12,16,18], while other studies have reported null [13,19] or even a positive [15] correlations. 

Moreover, previous evidence suggests that protein actions on GDM may vary by food sources. Instead of focusing on individual foods, evaluating the effects of dietary protein patterns is important because dietary patterns reflect food intake over a period and the cumulative effects of multiple nutrients [20]. It was reported that dietary patterns characterized by high intakes of processed and red meat, sweets, and snacks were related to an increased risk of GDM [16,18,19,21,22,23], while plant-based or prudent patterns with an abundant intake of vegetables, fruits, legumes, and deep-sea fish were related to a lower GDM risk [22,24,25,26]. However, there is no study exploring the relation between dietary protein and GDM risk from a whole-diet perspective. It remains unclear whether dietary protein patterns are differentially related to GDM. 

As suggested, the willingness of people to modify their dietary behavior might be influenced by their dietary patterns of protein intake [27], and the current dietary guidelines emphasize dietary patterns in the prevention of chronic diseases [28,29]. Therefore, an understanding of the role of dietary protein patterns on GDM development may have important public health implications. The objective of our study is to identify maternal dietary protein patterns during pregnancy and investigate whether the patterns are associated with the risk of GDM.

## 2. Materials and Methods

### 2.1. Study Design and Population

The data were obtained from the baseline investigation of a prospective GDM cohort study (ClinicalTrial.gov number: NCT03023293) conducted in Guangzhou, China. Pregnant women aged 20–45 years were recruited at 20 to 28 weeks of gestation during 2017–2018. We excluded subjects diagnosed with preexisting diabetes mellitus, cardiovascular disease, thyroid disease, hematopathy, polycystic ovary syndrome, mental-health disorders before pregnancy, and women with an infection during pregnancy or multiple gestation. 

A total of 1035 pregnant women were included. We further excluded those who reported implausible daily energy intake estimates of above 4000 kcal or below 800 kcal (n = 21). Thus, altogether,1014 women were included in the final analysis. Furthermore, this study obtained informed consent from all the subjects at initial enrollment and was approved by the Ethics Committee of the School of Public Health at Sun Yat-Sen University. 

### 2.2. Dietary Assessment

Dietary information during the past month before recruitment was assessed using a certified food frequency questionnaire (FFQ) and completed during a face-to-face interview.The FFQ consisted of 81 food items, covering the most common foods consumed in China, and was shown to be valid and reproducible among Chinese women in Guangzhou [30]. Subjects were required to recall their consumption frequency for each food item (number of times per day, week, or month) and food portion size. The food intake per frequency was presented in natural units (e.g., 1 egg), grams (e.g., 100 g of cooked chicken), or household measures (e.g., 1 spoon). A photo booklet with standard food portion sizes was provided to help participants estimate their intake portion for each food item.

Food consumption was converted into nutrient intakes according to the 2004 Chinese Food Composition Table [31]. We also investigated the consumption of nutrient supplement and computed nutrients according to the manufacturer instructions. Dietary nutrient intake was adjusted for total energy intake using the regression residual method [32]. The daily intake of protein from 81 food items was calculated in g/day and assigned into 10 mutually exclusive food groups according to the 2016 Chinese Dietary Guidelines [28]. The 10 food groups included grains, vegetables, beans, fruits, red meat, poultry, dairy, aquatic products, eggs, and nuts and seeds. AK-means cluster analysis was conducted to calculate the percentage contribution of total protein intake [protein from specific food group (g/day)/total protein intake (g/day) × 100].

### 2.3. Assessment of GDM

All of the participants were routinely scheduled for a 75g oral glucose tolerance test (OGTT) between 20 and 28 weeks of gestation. Maternal plasma glucose levels during OGTT, including fasting plasma glucose (FPG) and OGTT 1 h and 2 h glucose (postprandial glucose), were measured in a standardized clinical laboratory via an automatic biochemical analyzer (ARCHITECT i2000SR; Abbott Diagnostics, Chicago, IL, USA). GDM was diagnosed according to the OGTT if they met at least one of the following criteria [33]: FPG ≥ 5.10 mmol/L; OGTT 1-h plasma glucose ≥ 10.00 mmol/L; or OGTT 2-h plasma glucose ≥ 8.50 mmol/L.

### 2.4. Assessment of Covariates

We collected data on socio-demographic characteristics (maternal age, monthly household income, and educational level), lifestyle factors (alcohol use and smoking status) during pregnancy, history of GDM, and family history of diabetes via self-reported questionnaires. Maternal educational level was classified into 4 groups (senior high school and below, high or technical secondary school, junior college and college, and postgraduate and above). Monthly household income was categorized into 4 groups (≤4000, 4001–6000, 6001–10,000, and >10,000 RMB). History of GDM was divided into 3 groups (yes, no, and nulliparous). The data for the intensity of maternal physical activity were collected using the International Physical Activity Questionnaire [34] and were expressed in metabolic equivalents (METs). 

Information on pre-pregnancy weight and gestational age was abstracted from medical records. Maternal height was measured by trained nurses. Pre-pregnancy body mass index (BMI) was calculated as pre-pregnancy weight (kg) divided by the square of height (m). The subjects were divided into two weight groups according to the Chinese criteria for adults [35]: underweight or normal (<23.9 kg/m^2^) and overweight or obese (≥24.0 kg/m^2^).

### 2.5. Statistical Analysis

Maternal characteristics and dietary consumption were, respectively, described as means ± standard deviation (SD) or numbers and percentages (%) for continuous variables and categorical variables. The differences in the basic characteristics of participants with and without GDM were determined by the t-test or Chi-square test. 

Dietary protein patterns were constructed using the K-means cluster approach according to protein-rich food groups. Previous studies have reported a reasonable reproducibility and fair to modest validity of the dietary patterns derived by cluster analysis [36]. Firstly, the percentage of total dietary protein provided by each food group was calculated for each individual. Due to the cluster analysis being sensitive to outliers, we excluded subjects whose protein contribution was 5 SDs away from the mean protein contribution for each group and verified each food group contributing more than 0.5% of the total daily protein. Secondly, we used the FASTCLUS procedure in SAS software to generate dietary protein clusters. It is required that the number of clusters (k) be specified before analysis. In our study, the procedure was performed with pre-determined numbers of clusters (3–5 times) to assess the optimal number of clusters representing the dietary protein patterns in the current sample. Finally, the three-cluster set was applied because it distinguished the most meaningful separated dietary clusters; additionally, subjects were distributed well between three clusters, presenting a high F ratio.

We estimated the associations between dietary protein intake and protein patterns with risk of GDM using logistic regression. Potential confounders were adjusted in the logistic regression. We adjusted for gestational age, pre-pregnancy BMI, as well as age in model 1. In model 2, we additionally adjusted for family history of diabetes and history of GDM. Model 3 was further adjusted for smoking status, alcohol use during pregnancy, physical activities, dietary energy intake, protein-to-energy ratio, carbohydrate-to-energy ratio, fat-to-energy ratio, fiber, and cholesterol. In the last model, educational level and monthly household income were further adjusted. The total animal and plant protein intakes were divided into quartiles. The relation of the quartiles of the maternal protein intake to GDM risk were also tested using logistic regression analysis. We conducted linear trend tests across the quartiles of protein intake by taking the median of each quartile as continuous variables. All the analyses were conducted with SAS 9.4 (SAS Institute Inc., Cary, NC, USA). We considered *p* < 0.05 in the two-sided test as significant.

## 3. Results

### 3.1. Characteristics of the Participants

The general characteristics of the participants are presented in Table 1. Among the 1014 pregnant women, 191 (18.84%) were GDM. Pregnant women with GDM, compared to non-GDM, were more likely to have a higher age, pre-pregnancy BMI, and lower physical activity levels. Furthermore, GDM patients also had a higher percentage of advanced age (27.75% vs. 10.64%), overweight or obese (20.44% vs. 13.26%), and history of GDM (8.42% vs. 1.60%). No statistical differences were examined for the other socioeconomic characteristics of the two groups. 

The characteristics according to dietary protein patterns are presented in Appendix A. Women with the plant–dairy–eggs protein pattern had a lower gestational age, a higher percentage of advanced age, and a higher pre-pregnancy BMI, with no significant differences for the other characteristics.

### 3.2. Dietary Intake Characteristics

Table 2 summarizes the dietary consumption of energy-adjusted nutrients by GDM status. Women with GDM consumed more cholesterol, total protein, energy from total protein, animal protein, and protein from aquatic products. There were no significant differences between the other dietary nutrient intakes of the two groups.

According to the results of K-means cluster analysis, we derived three dietary protein patterns, namely, plant–dairy–eggs protein, white meat protein, and red meat protein patterns. The prevalence of GDM in these three groups was 16.23%, 20.88%, and 19.28%, respectively (*p* >0.05) (Appendix A). The average percentage of the total protein intake from the participant food groups in three dietary protein patterns is presented in Figure 1 and Appendix A. Women in the plant–dairy–eggs protein pattern (n = 302) consumed a relatively higher amount of protein from beans (7.17%,), vegetables (8.19%), fruits (3.62%), nuts and seeds (6.65%), dairy (13.41%), and eggs (9.97%). The white meat protein pattern (n = 297) was characterized by a relatively larger protein intake from aquatic products (14.56%) and poultry (11.18%). The red meat protein pattern (n = 415) consisted of a higher protein intake from red meat (31.50%). The analysis of variance showed significant differences among the percentage contributions of protein intake from each food group across the three dietary pattern groups (*p* < 0.05, Appendix A). 

Daily nutrient consumption across the dietary protein patterns are presented in Appendix A. Women with the plant–dairy–eggs protein pattern had a higher consumption of fiber, polyunsaturated fatty acids, carbohydrates, and plant protein and a lower consumption of total energy, saturated fatty acids, monounsaturated fatty acids, fat, and total protein, with no significant differences observed for cholesterol intake. The dietary consumption of the food groups was significantly different across the dietary protein patterns except for grain and nut and seed intake (Appendix A). 

### 3.3. Association between Dietary Protein Intake and Risk of GDM

As shown in Table 3, total and animal protein intakes were positively related to risk of GDM.The multivariate-adjusted ORs (95%) for GDM risk among the total population with the highest, as compared with the lowest, quartiles of intakes were 6.27 (95% 1.71–23.03) for total protein and 5.43 (95% CI 1.71–17.22) for animal protein intake. We suggest no significant association between plant protein intake and GDM risk. 

### 3.4. Association between Dietary Protein Patterns and Risk of GDM

As shown in Table 4, compared to women with the plant–dairy–eggs protein pattern, women with white meat protein or red meat protein patterns had higher risks of GDM, and the ORs (95% CI) were 1.83 (1.04–3.24) and 1.80 (1.06–3.07), respectively. In the stratified analysis, the associations above were more pronounced in overweight or obese women, but not in their counterparts (Appendix A), with no significant interaction. In the sensitivity analysis, compared to the red meat protein pattern, the plant–dairy–eggs protein pattern was related to a lower risk of GDM (OR: 0.56; 95%CI: 0.33–0.95), while no statistical association was found for the white meat protein pattern (Appendix A). When compared to women with the white meat protein pattern, women with the plant–dairy–eggs protein pattern had a lower risk of GDM (OR: 0.55; 95%CI: 0.31–0.96), while no significant association was observed between the red meat protein pattern and GDM risk (Appendix A).

## 4. Discussion

In this study, we observed that total and animal protein intakes during pregnancy were positively associated with GDM risk. Furthermore, we firstly identified three dietary protein patterns in pregnant women, which were labeled the plant–dairy–eggs protein pattern, the white meat protein pattern, and the red meat protein pattern. Compared with the plant–dairy–eggs protein pattern, both the white meat protein pattern and the red meat protein pattern were associated with higher GDM risks. 

Our findings support the positive associations between total and animal protein intakes with GDM risks, which is consistent with previous studies [12,13,14,15,16]. Liang et al. [14] demonstrated that higher dietary intakes of total and animal protein during mid-pregnancy were associated with an increased risk of GDM in pregnant women in Southwest China. The results from a cross-sectional study in Singapore and a prospective cohort study in Hubei, China, have also led to similar conclusions [15,16]. Furthermore, we observed a non-significant relationship between plant protein intake and GDM risk. This is consistent with several previous studies [13,14,16,19], whereas other studies have reported a negative [12,18,37], or even a positive [15], correlation, which may be explained by the discrepancies in race/ethnicity of the study populations [22].

Due to the distinct actions of dietary protein from different food sources on GDM risk, it is necessary to evaluate the association between dietary protein from a whole-diet perspective. We observed that women with the red meat protein pattern had a higher risk of GDM than women with the plant–dairy–eggs protein pattern. Although there is no previous study on the association of dietary protein pattern and GDM risk in pregnant women, this novel finding may give support to previous studies on the excess consumption of red meat. A recent meta-analysis demonstrated that red meat intake was positively associated with GDM risk (pooled RR: 1.72; 95%CI: 1.48–2.00) [38]. Liang et al. [14] also indicated that women in mid-pregnancy with a higher meat consumption had an increased risk of GDM, while intakes of beans and nuts were not significantly associated with GDM risk. For pre-pregnancy dietary patterns, previous studies have also suggested that a Western dietary pattern [17,39], characterized by a high consumption of meat-based products and processed foods or higher red meat intakes [12,17,40], were related to a significantly elevated risk of GDM. Bao et al. [12] also found that the substitution of nuts and legumes for red meat was associated with a lower risk of GDM. In the present study, the participants consumed a higher proportion of protein from red meat (16.66 g/d) than other protein sources, especially in women with the red meat protein pattern (23.06 g/d). Therefore, moderately decreasing the proportion of protein from red meat should be taken as a dietary suggestion for pregnant women for GDM prevention.

The findings we observed were biologically plausible, although the underlying mechanisms are still unclear. Accumulating epidemiological evidence has revealed that heme iron and branched-chain amino acids (BCAAs) in red meat might lead to the development of insulin resistance and increase the risk for GDM [40,41,42]. On one hand, the absorption of heme iron is 10–15 times higher than that of non-heme iron, and heme iron is more likely to lead to an abundance of body iron storage [43]. An overload of iron plays an essential role in diabetes pathogenesis, mediated both by pancreatic β-cell failure and insulin resistance [44,45]. On the other hand, metabolomics studies have demonstrated that BCAAs and aromatic amino acids had highly significant associations with future diabetes [46,47]. Considering that an increase in the consumption red meat may increase plasma concentration of BCAAs [48], the association between the red meat protein pattern and GDM may be partly explained by the detrimental effect of BCAAs.

Furthermore, pregnant women with the white meat protein pattern, characterized by higher percentage of total protein from poultry and aquatic products, had a higher risk of GDM than those with the plant–dairy–eggs protein pattern. Similarly, Du et al. [26] found that the dietary pattern characterized by baked/fried food and white meat was associated with an increased risk of GDM (aOR: 4.40, 95% CI: 1.58–12.22, *p* < 0.05) when compared with the prudent pattern. Pang et al. [15] reported a positive association between protein intake from aquatic products and GDM risk. Although a non-significant association between poultry and aquatic products intake with GDM risk was also reported [12,14,15,40], several studies have presented a positive association between dietary poultry [49] and seafood [15,50] intakes and T2DM risk. Fan et al. [51] further confirmed a positive dose-response relationship between poultry intake and T2DM risk. A meta-analysis also elucidated that a higher intake of fish (more than 105 g/week) was positively associated with risk of T2DM [52]. Diets rich in fish affect the circulating concentration of trimethylamine N-oxide (TMAO) [53], which is associated with a higher risk of diabetes [54,55,56]. In the current study, women with the white meat protein pattern consumed much more poultry (48.2 g/d) and aquatic products (72.9 g/d) than women with the plant–dairy–eggs protein pattern (13.4 g/d and 30.5 g/d). Another major common characteristic of the red meat and white meat protein patterns is the higher level of total fat and animal fat than that in plant–dairy–eggs protein pattern. In our study, total fat intake was significantly higher in women with the red meat (37.60% of total energy intake) and white meat (37.76% of total energy intake) protein patterns, and was far beyond the recommended amounts (20–35% of total energy intake) [28,29,57].

On the other hand, although several studies have indicated that dairy and egg intakes were associated with an increased risk of GDM [9,14,56], the dose-effect meta-analysis elucidated that low-fat dairy and eggs consumed in quantities less than 50g per day may be protective against diabetes [58,59,60]. In the current study, the mean consumption of eggs is 38.96g per day, which is in line with the dietary recommendation for pregnant women [28]. Apart from being good sources of plant protein for beans and nuts, beans are rich in antioxidant vitamins and minerals and other phytochemicals with antioxidant, anti-inflammatory, and antimicrobial properties [61,62]. Nuts are also abundant in fatty acids, fiber, and magnesium [63], which are related to a reduced risk of diabetes [64,65]. With the addition of the beneficial effects of vegetables and fruits on GDM [16,39], the plant–dairy–eggs protein pattern may be considered a healthy dietary pattern for pregnant women.

This is the first study to investigate the associations between dietary protein patterns and GDM risk in pregnant women. We used cluster analysis to derive the dietary patterns based on protein-rich foods to identify distinct and relatively homogenous groups [66] and evaluate their associations with risk of GDM. Our findings can enhance the conceptual understanding of protein actions on health and provide evidence to support the prevailing dietary recommendation of a balanced diet with limited animal protein [28]. Pregnant women should give priority to the plant–dairy–eggs protein pattern and avoid excessive consumption of red and white meat to minimize their risk of GDM.

Several potential limitations should be acknowledged. Firstly, the cross-sectional design of this study may limit the cause–effect relationship. However, the dietary assessments were conducted before the women were informed of GDM diagnosis, which can partly rule out the possibility of reverse causation. Secondly, recall bias is inevitable in dietary assessments with the FFQ. Nonetheless, the validated semi-quantitative FFQ [30] might better reflect the habitual dietary intake over a period of time. Trained investigators also used photographs of the food with standard portion sizes for assistance in the face-to-face interviews, which minimized the bias. Finally, our study population primarily consisted of Han Chinese in Guangzhou, China, which may limit the generalization of the findings to other populations. However, the homogeneity of our population may also help reduce unmeasured confounding by socioeconomic status.

## 5. Conclusions

Higher dietary intakes of total and animal protein during mid-pregnancy were associated with an increased risk of GDM. Furthermore, compared with women who had the plant–dairy–eggs protein pattern, women with the red meat or white meat protein patterns had a higher risk of GDM. Pregnant women should follow a balanced diet and avoid excessive consumption of meat-based protein.

## Figures and Tables

**Figure 1 nutrients-14-01623-f001:**
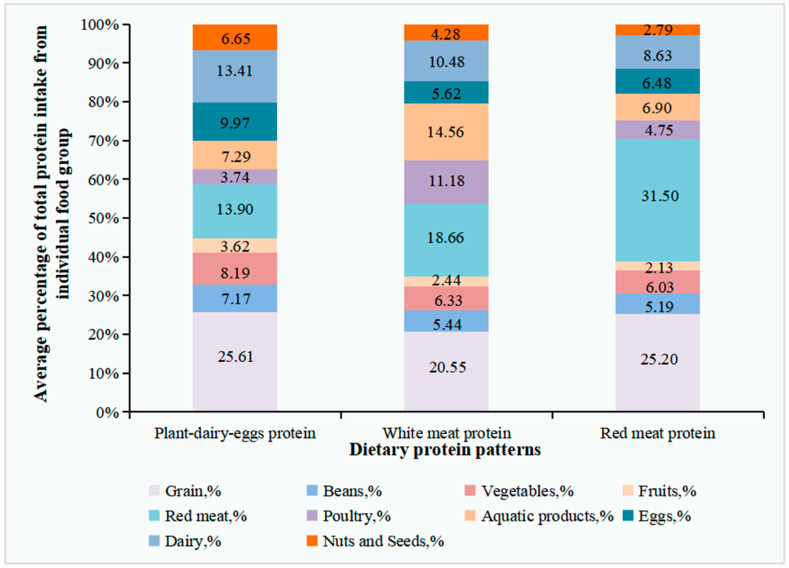
Average percentage of total dietary protein intake from individual food groups across the protein food cluster analysis. A K-means cluster analysis was performed to categorize subjects into mutually exclusive groups. The name of the clusters was determined according to the percentage that represented the highest consumption of one or two food groups. Percentage of total protein intake across each food group was used.The analysis of variance showed that there were significant differences among the percentage contributions of protein intake from each food group in the three dietary pattern groups (*p* < 0.05) (Appendix A).

**Table 1 nutrients-14-01623-t001:** Characteristics of pregnant women by categories of GDM.

Characteristics	Total	GDM	Normal	*p*
	(n =1014)	(n = 191)	(n = 823)	
Age, y	30.05 ± 4.84	31.99 ± 5.07	29.60 ± 4.68	**<0.001**
<35, y	851 (84.01)	138 (72.25)	687 (89.22)	**<0.001**
≥35, y	162 (15.99)	53 (27.75)	82 (10.64)	
Gestational age, week	25.45 ± 2.25	25.24 ± 2.47	25.50 ± 2.29	0.178
Pre-pregnancy BMI, kg/m^2^	20.57 ± 2.81	21.43 ± 3.47	20.38 ± 2.72	**<0.001**
Overweight or obese, n (%)	120 (12.62)	37 (20.44)	109 (13.26)	**<0.001**
Underweight or normal, n (%)	831 (87.38)	144 (79.56)	712 (86.72)	
Smoking, yes, n (%)	44 (4.37)	10 (5.24)	34 (4.16)	0.515
Alcohol use, yes, n (%)	35 (3.47)	6 (3.14)	29 (3.55)	0.782
Physical activity, METs·h/w	31.72 ± 27.39	27.70 ± 21.95	32.65 ± 28.43	**0.024**
Family history of diabetes, yes, n (%)	150 (14.90)	32 (16.75)	118 (14.46)	0.427
History of GDM, n (%)				**<0.001**
Yes	29 (2.90)	16 (8.42)	13 (1.60)	
No	585 (58.44)	113 (59.47)	472 (57.20)	
Nulliparous	387 (38.66)	61 (32.11)	326 (40.20)	
Educational level, n (%)				0.577
Senior high school and below	183 (18.48)	33 (17.37)	150 (18.75)	
High or technical secondary school	213 (21.52)	44 (23.16)	169 (21.13)	
Junior college and college	534 (53.94)	98 (51.58)	436 (54.50)	
Postgraduate and above	60 (6.06)	15 (7.89)	45 (5.63)	
Monthly household income, n (%)				0.927
≤4000 RMB	209 (21.28)	38 (20.32)	171 (21.51)	
4001–6000 RMB	236 (24.03)	44 (23.53)	191 (24.15)	
6001–10,000 RMB	243 (24.75)	50 (26.74)	193 (24.28)	
>10,000 RMB	294 (29.94)	55 (29.41)	239 (30.06)	

Values are shown with mean ± standard deviation, or numbers and proportions. The bold values indicated that there were statistical significance (*p* < 0.05).

**Table 2 nutrients-14-01623-t002:** Daily nutrient consumption of pregnant women by categories of GDM.

Nutrients	Total	GDM	Normal	*p*
	(n =1014)	(n = 191)	(n = 823)	
Total energy, kcal/day	1803.15 ± 496.19	1823.79 ± 479.81	1798.36 ± 504.01	0.531
Saturated fatty acids, g/day	19.92 ± 4.02	19.92 ± 4.50	19.99 ± 4.00	0.709
Monounsaturated fatty acids, g/day	27.78 ± 5.63	27.93 ± 5.79	27.86 ± 5.82	0.937
Polyunsaturated fatty acids, g/day	20.96 ± 5.75	20.39 ± 5.74	21.03 ± 5.77	0.195
Cholesterol, mg/day	404.03 ± 161.29	463.54 ± 172.78	397.78 ± 159.64	**0.004**
Fiber, g/day	11.10 ± 3.09	11.17 ± 3.27	11.01 ± 3.04	0.516
Carbohydrates, g/day	217.98 ± 31.01	216.04 ± 32.97	218.11 ± 31.07	0.411
% Energy	48.14 ± 6.66	47.57 ± 6.85	48.19 ± 6.67	0.246
Fat, g/day	73.95 ± 11.63	74.27 ± 12.52	74.22 ± 11.80	0.960
% Energy	37.41 ± 5.76	37.60 ± 6.09	37.46 ± 5.76	0.770
Protein, g/day	71.37 ± 11.30	73.14 ± 11.31	71.06 ± 11.33	**0.022**
% Energy	15.63 ± 2.59	16.05 ± 2.62	15.54 ± 2.58	**0.015**
Animal protein, g/day	40.83 ± 13.48	43.12 ± 13.65	40.38 ± 13.49	**0.018**
Plant protein, g/day	30.56 ± 5.55	30.07 ± 5.96	30.72 ± 5.52	0.184
Protein sources				
From grain, g/day	16.15 ± 4.65	15.82 ± 4.99	16.22 ± 4.56	0.338
From beans, g/day	4.18 ± 3.59	4.04 ± 3.97	4.24 ± 3.50	0.507
From vegetables, g/day	4.71 ± 2.32	4.83 ± 2.39	4.55 ± 2.29	0.112
From fruits, g/day	1.82 ± 1.07	1.85 ± 1.13	1.79 ± 1.04	0.536
From red meat, g/day	16.66 ± 9.69	17.40 ± 10.04	16.54 ± 9.76	0.321
From poultry, g/day	4.66 ± 4.04	5.10 ± 4.34	4.61 ± 3.97	0.135
From aquatic products, g/day	6.93 ± 6.44	7.83 ± 6.56	6.68 ± 6.42	**0.032**
From eggs, g/day	5.04 ± 3.31	5.42 ± 3.46	4.90 ± 3.28	0.054
From dairy, g/day	7.57 ± 5.11	7.35 ± 5.26	7.66 ± 5.05	0.421
From nuts and seeds, g/day	3.34 ± 4.12	3.07 ± 3.72	3.48 ± 4.21	0.247

Values are presented as mean± standard deviation. Energy-adjusted intake estimated by the residual method. The bold values indicated that there were statistical significance (*p* < 0.05).

**Table 3 nutrients-14-01623-t003:** The association between maternal protein intake and GDM risk.

	Energy-Adjusted Total Protein Intake Quartiles, OR (95%CI)				*p*-Trend
	Q1	Q2	Q3	Q4	
Total Protein	1	1.75 (0.90–3.44)	2.88 (1.20–6.91)	6.27 (1.71–23.03)	**0.017**
Animal Protein	1	1.91 (0.97–3.73)	3.04 (1.33–6.95)	5.43 (1.71–17.22)	**0.011**
Plant Protein	1	1.05 (0.61–1.83)	0.87 (0.46–1.66)	0.93 (0.38–2.25)	0.715

The model was adjusted for age, pre-pregnancy BMI, gestational age, family history of diabetes, GDM in a previous pregnancy, smoking status, alcohol use during pregnancy, physical activities, dietary energy intake, protein-to-energy ratio, carbohydrate-to-energy ratio, fat-to-energy ratio, fiber, cholesterol, educational level, andmonthly household income. The bold values indicated that there were statistical significance (*p* < 0.05).

**Table 4 nutrients-14-01623-t004:** The odds ratios of GDM across three dietary patterns.

Model	Dietary Protein Patterns		
	Plant–Dairy–Eggs	White Meat	Red Meat
GDM (N, %)	49 (16.23%)	62 (20.88%)	80 (19.28%)
Unadjusted OR (95% CI)	1.00	1.36 (0.90–2.06)	1.23 (0.83–1.82)
Adjusted OR (95% CI)			
Model 1	1.00	**1.80 (1.13–2.85)**	1.52 (0.99–2.35)
Model 2	1.00	**1.82 (1.14–2.90)**	**1.59 (1.02–2.46)**
Model 3	1.00	**1.96 (1.12–3.34)**	**1.84 (1.09–3.10)**
Model 4	1.00	**1.83 (1.04–3.24)**	**1.80 (1.06–3.07)**

Model 1 was adjusted for gestational age, age, pre-pregnancy BMI; Model 2 was further adjusted for family history of diabetes and GDM in a previous pregnancy; Model 3 was further adjusted for smoking status, alcohol use during pregnancy, physical activities, dietary energy intake, protein-to-energy ratio, carbohydrate-to-energy ratio, fat-to-energy ratio, fiber, cholesterol; Model 4 was further adjusted for educational level and monthly household income. The bold values indicated that there were statistical significance (*p* < 0.05).

## Data Availability

The data presented in this study are available on request from the corresponding author. The data are not publicly available due to ethical requirements.

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
