# Peer review of "Dietary Protein Patterns during Pregnancy Are Associated with Risk of Gestational Diabetes Mellitus in Chinese Pregnant Women"

_nutrients, 2022, doi:10.3390/nu14081623_

Round 1

Reviewer 1 Report

In the beginning, I want to congratulate you on an interesting article. I appreciate the effort put into it.
I have just a few comments, they come down to simple fixes as below:

  • line 21 - please specify which food questionnaire you used,
  • line 213 - "CI is missing in the brackets
  • table 4 - please indicate that the numbers refers to OR/aOR/CI 
  • lines 297-299 - please give statistical significance 
  • line 304 - what was the fat intake in plant-dairy-eggs pattern?
  • in my opinion the clusters should be described in more detail

Thank you 

Reviewer 2 Report

Aim of this cross-sectional study is to investigate maternal dietary protein patterns during pregnancy and identify whether the patterns are associated with the risk of gestational diabetes mellitus. 

This study addresses a very important issue because nutrition during pregnancy is central to reducing the risk of diabetes and subsequent health risks for the unborn child. 

The authors conducted the study in a formally impeccable manner. 
There are a few minor criticisms to be addressed before publication.

  1. Why did the authors combine plant protien with egg and dairy protein? Assessing the differences between these would have been interesting. As well as differentiating the two patterns of plant and animal proteins.
  2. it would also be useful to assess the risks of diabetes differently in relation to protein intake for overweight, obese, noemai and underweight individuals. 
  3. Recent studies have correlated TMAO with the risk of diabetes. This could explain the increased risk of fish protein in these subjects. The authors could comment on this.
